# Adaptive Trait Variation in Seedlings of Rare Endemic Mexican Spruce Provenances under Nursery Conditions

Christian Wehenkel [1,*], José Marcos Torres-Valverde [2], José Ciro Hernández-Díaz [1], Eduardo Mendoza-Maya [3], Artemio Carrillo-Parra [1], Santiago Solis-González [4] and Javier López-Upton [5]

1 Instituto de Silvicultura e Industria de la Madera, Universidad Juárez del Estado de Durango, Durango 34120, Mexico

2 Maestría Institucional en Ciencias Agropecuarias y Forestales, Universidad Juárez del Estado de Durango, Durango 34000, Mexico

3 Programa Institucional de Doctorado en Ciencias Agropecuarias y Forestales, Universidad Juárez del Estado de Durango, Durango 34000, Mexico

4 Instituto Tecnológico de El Salto, Durango 34942, Mexico

5 Postgrado en Ciencias Forestales Colegio de Postgraduados, Montecillo, Texcoco 56264, Mexico

* Correspondence: wehenkel@ujed.mx; Tel.: +52-(618)-102-0873

**Abstract:** The distributions of the three Mexican spruces are fragmented, possibly leading to phenological, morphological and genetic differentiation, which is partly caused by local adaptation. In this study, we estimated for the first time the intra- and inter-specific phenotypic variation in 5641 seedlings from provenances of the three Mexican spruces. We examined (i) provenance-related differences in the seedling survival rate, diameter ($D$), height ($H$) and seed weight ($SW$) as quantitative traits, (ii) the association between the survival rate, $D$, $H$ and $SW$ and climatic and soil variables in the *Picea* provenances and (iii) (narrow-sense) heritability (within-provenance) based on $D$ and $H$ under the same nursery conditions, assuming that the response can be considered as a proxy for quantitative genetic differentiation between provenances. All Mexican spruce species differed significantly in $H$, and all eight provenances studied were significantly different in $D$ and $H$, except for two neighboring provenances of *P. mexicana*. Very strong, significant correlations (up to $R^2 = 0.96$) were found between $H$, the survival rate and $SW$ with respect to environmental factors of provenance/seed origin. Additionally, the heritability index explained a high percentage of the provenance-related variance. The use of germplasm for restoration in different sites and with different populations requires collecting seeds from numerous trees from as many provenances as possible, but should be carried out with caution owing to the apparently strong local adaptation in provenances of the Mexican spruces.

**Keywords:** Picea; Mexican spruces; quantitative genetic traits; genetic differentiation; provenance; heritability; nursery conditions; trans-specific trait; tree species conservation

## 1. Introduction

Provenance in forestry can be defined as the particular place where trees are growing naturally or the place of origin of any germplasm. The term provenance is generally applied to a tree population growing in a certain location and not to an individual tree [1]. Although the concept of "provenance" is commonly used at the species level, Gregorius et al. [2] developed an operational trait concept that, under specific conditions, allows for the measurement of variation in traits that can be observed in members of the same species and also in members of different species. This concept is based on research methods that can be adapted to a broader range of organisms without altering their traits. These traits, referred to as "trans-specific" traits, have been measured in several previous studies, e.g., [3–5]. Provenance–progeny trials enable the study of the genetic basis of quantitative traits and assessment of the degree of local adaptation [6–10]. In this context, heritability can be defined as the proportion of variance in phenotypic traits that can be attributed

to genetic variation in a provenance [11]. Several provenance trials have reported the heritability of quantitative traits in forest trees, such as tree height and diameter [12] and seed size [13–17].

Provenance trials are time-consuming and usually only provide a measure of phenotypic differences among populations under common conditions. However, no breeding experience is available, and provenance trials have not been carried out with the three rare, endemic and endangered Mexican spruces *Picea chihuahuana* Martínez, *P. martinezii* Patterson and *P. mexicana* Martínez (NOM-059-ECOL-2010, SEMARNAT, 2010) [18]. *P. chihuahuana* is only located in the Sierra Madre Occidental (SMOc) in the states of Chihuahua and Durango, Mexico (total cover ~300 ha). The 40 documented provenances of this species have between 21 and 5546 individuals (a total of approximately 42,600 mature trees), which are found in three geographical groups approximately 300 km apart [19,20]. Growing in an area of approximately 157 ha, *P. martinezii* only occurs in the Sierra Madre Oriental (SMOr) at elevations between 1800 and 2500 m in the four documented provenances in the state of Nuevo León, Mexico [18]. By contrast, *P. mexicana* occurs in three montane to subalpine sites above 3000 m, thriving in an area of approximately 173 ha; two provenances are in two of the highest peaks in the SMOr, at the border of the Mexican states of Coahuila and Nuevo León, and the third location is the highest peak in the SMOc, in the state of Chihuahua [21].

Given the high degree of isolation of the provenances of the three Mexican spruces (especially the isolation between these species), strong genetic differentiation and, respectively, a large genetic variation among the provenances have been observed and partly attributed to geographical (e.g., distance, relief, elevation), ecological and possibly pre- or post-zygotic barriers and time. Plastid, mitochondrial and nuclear sequences show that *P. mexicana* (a sister species of the more widespread *P. pungens* Engelm.), *P. chihuahuana* and *P. martinezii* are genetically distinct [22]. Moreover, isozyme analysis showed that the mean genetic differentiation (measured by the Fixation index, $F_{ST}$) for ten provenances of *P. chihuahuana* was 0.248 (i.e., 75.2% of the total genetic variation is within provenances and 24.8% among provenances), which is outside the range reported in most previous studies in conifers [23]. By contrast, the $F_{ST}$ was 0.024 for two provenances of *P. martinezii* [24] and 0.069 for the three provenances of *P. mexicana* [25], i.e., higher than the average $F_{ST}$ for spruces [23]. However, with SNP data, the $F_{ST}$ was 0.027 for the four provenances of *P. martinezii* and 0.028 for the three provenances of *P. mexicana* [18].

Some of the genetic differentiation among these provenances may also have resulted from adaptation to the local climate and soil [20]. In this context, climatic variation across mountains often results in local adaptation to elevation, e.g., [26,27]; as it is closely correlated with key climatic factors, such as temperature and aridity [28]. Knowledge of possible local adaptation could be helpful in ex situ conservation programs for these three Mexican spruces. For example, as many genotypes as possible should be used for restoration and assisted migration to cope with climate change, but these genotypes should also be sufficiently adapted to the new locations [29,30].

In the present study, we estimated the intra- and inter-specific phenotypic variation in seedlings from eight provenances of the three Mexican spruces by applying the transspecific trait concept of Gregorius et al. [2]. More specifically, we examined (i) provenance-related differences in the seedling diameter (*D*), height (*H*), survival rate and weight of 1000 seeds of parental trees (*SW*) as quantitative traits, (ii) the association between the survival rate, *D*, *H* and *SW* and climatic and soil variables in the *Picea* provenances and (iii) (narrow-sense) heritability (within-provenance) based on *D* and *H* in a common garden experiment under the assumption that the response can be considered as a proxy for quantitative genetic differentiation between provenances.

## 2. Materials and Methods

### 2.1. Study Site

The study was conducted between May 2019 and May 2020 in a nursery located in El Salto, Pueblo Nuevo, state of Durango, Mexico (23°47′11.72″ N, 105°21′55.18″ W) (https://www.worldweatheronline.com/el-salto-weather-averages/durango/mx.aspx, (accessed on 8 April 2023)) with seeds from eight provenances: the largest of *Picea chihuahuana* (Pch) in the central part of its distribution area, situated in Guanaceví, Durango (mean annual temperature [Mat], 10.7 °C, and precipitation [Map], 740 mm), the four documented provenances of *P. martinezii* (Pma) from Nuevo León (Mat, 13.5 °C, and Map, 788 mm) and the three reported populations of *P. mexicana* (Pme) located in two different states in Mexico: one in Chihuahua and two in Coahuila (Mat, 9.3 °C, and Map, 858 mm) [20,21]. For each provenance, 10–30 dominant trees were randomly chosen, and 50–250 cones were collected from each (Table 1 and Figure 1). Although only one provenance of Pch was included and the trial thus did not meet the requirements of a species-level provenance trial, this provenance was an asset in the genus-level trial. Moreover, the Pch seed weight and survival and diameter and height growth of Pch seedlings have never been reported in previous studies.

**Table 1.** Location and population size of the eight provenances of three species of Mexican spruces under study and number of trees sampled (*N*); data of *Picea chihuahuana* from [20] and of *P. martinezii* and *P. mexicana* from [21].

| Species | Code | Location Municipality, State | Population Size | Latitude (N) | Longitude (W) | Elevation (m) | *N* |
|---|---|---|---|---|---|---|---|
| *Picea chihuahuana* | Pch-QD | Quebrada de los Durán, Chiqueros, Guanaceví, Durango | 2628 | 26°08′48″ | 106°22′53″ | 2570 | 10 |
| *Picea martinezii* | Pma-AF | Agua Fría, Aramberri, Nuevo León | 2769 | 24°02′17″ | 99°42′39″ | 1820 | 20 |
| | Pma-AA | Agua Alardín, Aramberri, Nuevo León | 84,498 | 24°02′34″ | 99°44′04″ | 2120 | 30 |
| | Pma-EB | El Butano, Ejido La Trinidad, Montemorelos, Nuevo León | 1253 | 25°10′41″ | 100°07′37″ | 2180 | 30 |
| | Pma-LE | La Encantada, Ejido La Encantada, Zaragoza, Nuevo León | 712 | 23°53′24″ | 99°47′30″ | 2378 | 12 |
| *Picea mexicana* | Pme-EM | El Mohinora, Ejidos El Tule y Portugal, Guadalupe y Calvo, Chihuahua | 11,383 | 25°11′55″ | 100°21′52″ | 3113 | 30 |
| | Pme-LM | La Marta, Felipe de la Peña, Arteaga, Coahuila | 17,728 | 25°57′41″ | 107°02′32″ | 3494 | 29 |
| | Pme-EC | El Coahuilón, Ejido Nuncio, Arteaga, Coahuila | 2253 | 25°14′51″ | 100°21′17″ | 3528 | 28 |

### 2.2. Determination of Climatic Variables

The climatic model developed [31] on the basis of thin plate spline (TPS) [32,33] and applied to Mexico [34] was used to estimate the values of 20 climatic variables in each location of the eight provenances of Mexican spruces. The point estimates of climate measures were obtained from a database managed by the University of Idaho (https://charcoal2.cnre.vt.edu/climate/ (accessed on 8 April 2023)), for which the geographical coordinates (latitude, longitude and elevation) were required as input data to predict the climatic variables in each location. The minimum, maximum and mean values of these climate values are listed in Table 2.

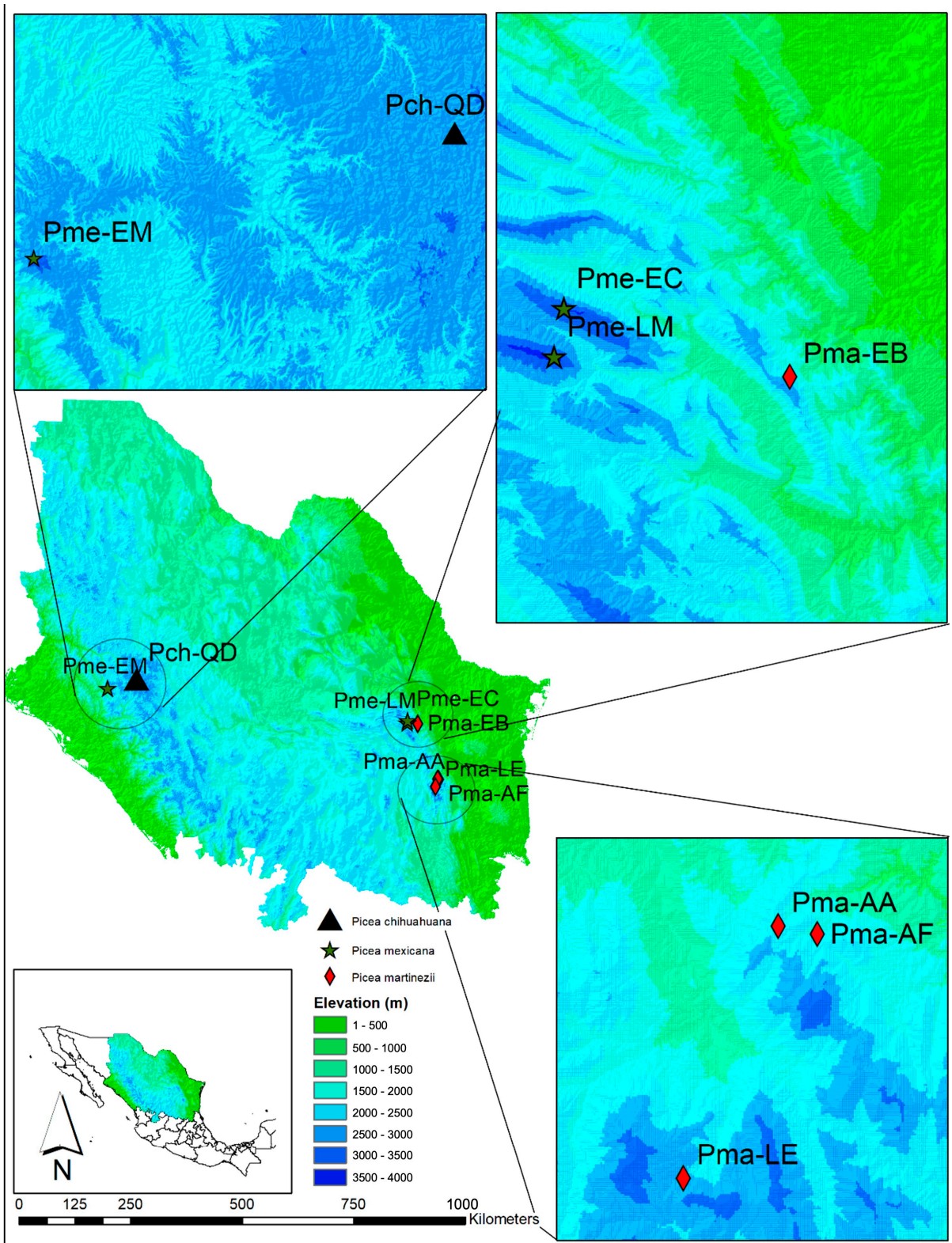

**Figure 1.** Locations of the eight provenances of the three Mexican spruces under study. *Picea chihuahuana* Quebrada de los Durán (Pch-QD); *P. martinezii* provenances of Agua Fría (Pma-AF), Agua Alardín (Pma-AA), El Butano (Pma-EB) and La Encantada (Pma-LE); *P. mexicana* provenances of El Mohinora (Pme-EM), La Marta (Pme-LM) and El Coahuilón (Pme-EC).

**Table 2.** Mean values of climate variable used for the environment–phenotype associations tests (* SE = standard error; data of *Picea chihuahuana* from [20] and of *P. martinezii and P. mexicana* from [21] for the time period 1961–1990).

| | Climatic Variable | Minimum | Maximum | Mean | SE * |
|---|---|---|---|---|---|
| Long | Longitude (degree) | −99.71 | −107.04 | −101.69 | 0.75 |
| Lat | Latitude (degrees) | 20.71 | 26.14 | 24.39 | 0.26 |
| Elev | Elevation (m) | 1820 | 3528 | 2644 | 223.00 |
| Mat | Mean annual temperature (°C) | 8.5 | 15.6 | 12.4 | 0.70 |
| Map | Mean annual precipitation (mm) | 649 | 1106 | 826 | 32.00 |
| Gsp | Growing season precipitation, April to September (mm) | 485 | 790 | 630 | 21.00 |
| Mtcm | Mean temperature in the coldest month (°C) | 3.7 | 11.6 | 8.5 | 0.80 |
| Mmin | Mean minimum temperature in the coldest month (°C) | −5.9 | 4.5 | 1.1 | 1.00 |
| Mtwm | Mean temperature in the warmest month (°C) | 12.5 | 18.5 | 15.4 | 0.80 |
| Mmax | Mean maximum temperature in the warmest month (°C) | 19.4 | 25.6 | 23.0 | 0.83 |
| Sday | Julian date of the last spring freezing (days) | 39 | 184 | 89 | 14.22 |
| Fday | Julian date of the first autumn freezing (days) | 268 | 348 | 302 | 11.29 |
| Ffp | Length of the frost-free period (days) | 74 | 298 | 200 | 25.93 |
| Dd5 | Degree-days above 5 °C | 1540 | 3873 | 2766 | 259.20 |
| Gsdd5 | Degree-days above 5 °C in the frost-free period | 489 | 3408 | 2058 | 339.60 |
| D100 | Julian date when the sum of degree-days above 5 °C reaches 100 | 15 | 71 | 31.5 | 4.38 |
| DD0 | Degree-days below 0 °C (based on mean monthly temperature) | 0 | 45 | 7.5 | 1.60 |
| Mminddo | Degree-days below 0 °C (based on mean minimum monthly temperature) | 29 | 1162 | 316 | 91.84 |
| Smrpb | Summer precipitation balance: (Jul + Aug + Sep)/(Apr + May + Jun) | 1.29 | 4.77 | 2.39 | 0.32 |
| Smrsprpb | Summer/Spring precipitation balance: (Jul + Aug)/(Apr + May) | 1.26 | 13.14 | 4.84 | 1.32 |
| Sprp | Spring precipitation (Apr + May) (mm) | 26 | 129 | 94 | 11.36 |
| Smrp | Summer precipitation (Jul + Aug) (mm) | 163 | 486 | 275 | 20.57 |
| Winp | Winter precipitation (Nov + Dec + Jan + Feb) (mm) | 76 | 228 | 113 | 8.88 |

The aforementioned climatic model estimates standardized monthly mean, minimum and maximum values of temperature and precipitation (e.g., mean annual precipitation (Map, mm), mean temperature of the warmest month (Mtwm, °C), Julian date of the first freezing date of autumn (Fday, days) and precipitation during the growing season (April-September; Gsp, mm), among others), which are based on data from weather stations in the northern states of Mexico, southern United States and the Caribbean countries for the period 1961–1990.

*2.3. Determination of Edaphic Variables*

In 2019, eight soil samples per provenance of *P. martinezii* and *P. mexicana* were collected to evaluate 27 edaphic variables (56 soil samples in total, 250 g each). The soil samples were randomly collected at a depth of 0–15 cm at the stem base of one spruce tree. Soil data for the *P. chihuahuana* provenance were taken from [20].

The following soil variables were considered: texture, density, concentration of calcium carbonate, pH and concentrations of potassium, magnesium, sodium, copper, iron, manganese, zinc and calcium in the soil, which were determined by the methods described by [35]. Phosphorus was determined by the method of [36]. Nitrate was determined by the method of [37], and the relative organic matter contents were measured by the method of [38]. Electrical conductivity was determined by the method described by [39]. Finally, the cation exchange capacity (CEC) was estimated on the basis of the ammonium acetate method (pH 8.5). The hydraulic conductivity was determined by the method of [40] and the percent of water saturation was measured by the method of [41]. These variables are listed in Table 3.

**Table 3.** Descriptive statistics for the 27 soil variables in provenance of *P. martinezii* and *P. mexicana* used for the environment–phenotype associations tests (* SE = standard error (sample = 56); data of *Picea chihuahuana* from [20] are not inserted in the table.).

| | Soil Variable | Minimum | Maximum | Mean | SE * |
|---|---|---|---|---|---|
| EC | Electrical conductivity (dS/m) | 0.47 | 1.07 | 0.84 | 0.07 |
| NO3 | Nitrate (kg/ha) | 86 | 628 | 261 | 63 |
| P | Phosphorus (mg/kg) | 9.76 | 75.74 | 37.47 | 7.11 |
| OM | Organic matter (%) | 5.2 | 24.7 | 13.5 | 2.6 |
| %CaCO$_3$ | Calcium carbonate (%) | 0 | 1.18 | 0.28 | 0.20 |
| %Sat | Percent saturation (%) | 48.0 | 104.2 | 70.7 | 6.7 |
| %Sand | Sand (%) | 22.2 | 61.4 | 44.1 | 5.6 |
| %Silt | Silt (%) | 25.3 | 58 | 38.7 | 4.1 |
| %Clay | Clay (%) | 7.6 | 26.6 | 17.1 | 2.1 |
| Den | Density (g/cm$^3$) | 0.54 | 7.80 | 1.66 | 0.93 |
| pH | pH | 4.59 | 7.62 | 6.06 | 0.42 |
| Ca | Calcium (mg/kg) | 180 | 10,624 | 4784 | 1129 |
| Mg | Magnesium (mg/kg) | 123 | 528 | 263 | 49 |
| Na | Sodium (mg/kg) | 40.8 | 86.5 | 56.5 | 2.6 |
| K | Potassium (mg/kg) | 666 | 2475 | 1574 | 163 |
| Fe | Iron (mg/kg) | 31.2 | 521 | 234 | 52 |
| Zn | Zinc (mg/kg) | 3.0 | 118.1 | 19.7 | 14.9 |
| Mn | Manganese (mg/kg) | 18.9 | 171.7 | 59.0 | 17.5 |
| Cu | Copper (mg/kg) | 0.16 | 4.67 | 3.31 | 0.25 |
| CEC | Cation exchange capacity (meq/100 g soil) | 11.4 | 56.7 | 27.7 | 5.8 |
| HC | Hydraulic conductivity (cm/h) | 3.7 | 137.1 | 29.7 | 16.3 |

*2.4. Weight of 1000 Seeds, Survival and Growth Rate of Seedlings*

The weight of 1000 seeds (*SW*, g) of each three randomly chosen parental trees per provenance was measured on a digital scale (Velab model No. VE-5000H, Mexico), and the mean weight was calculated for each provenance. We sowed 5641 seeds for evaluation of seedling survival and growth traits. A total of 93 seeds were from 10 *P. chihuahuana* trees; 2853 seeds were from 92 *P. martinezii* trees; and 2695 seeds were from 87 *P. mexicana* trees.

For common garden experiments, the seeds were first soaked in hydrogen peroxide (2%, *v/v*) for 24 h and were then sown in 170 cm$^3$ plastic tubes containing a mixture of 60% peat moss, 20% agrolite, 20% vermiculite and 3 kg·m$^3$ of a controlled-release fertilizer (Multicote®; 15% N + 7% P$_2$O$_5$ + 15% K$_2$O + 2 MgO + ME fertilizer). The tubes (containing seeds) were distributed in the 56 plastic trays, each with 98-hole plastic containers (7 × 14 cavities), in a randomized complete-block design. Provenance data-codes were maintained throughout the 12-month monitoring period; i.e., "species—provenance—seedling number—repetition".

Additionally, N-P-K Multicote® 08-25-17 soluble fertilizer (5.0%, *v/v*) was added once a week during two months of early growth of the seedlings. During the rapid growth phase, N-P-K Multicote® 20-09-20 soluble fertilizer (20%, *v/v*) was added once a week until six months of age. Finally, at the hardening stage, N-P-K Multicote® 04-25-40 soluble fertilizer (5.0%, *v/v*) was applied once a week until 12 months of age. Water was supplied by an automatic irrigation system three days/week. During the germination and growth of seedlings (first six months), a 720-gauge polyethylene plastic cover was used to protect the seedlings from ultraviolet rays, with no shading mesh. At the preconditioning stage (2 months), the seedlings were kept under a 50% light mesh shade, whereas, at the hardening stage (from month seven) the seedlings were kept outdoors.

Differences in growth of seedlings were evaluated monthly over a 12-month period by measuring basal diameter (at plant collar) (*D*, mm) with a digital Vernier scale of resolution tenths of a millimeter (AVEDISTANT, LCD6). The plant height at the apical growth bud (*H*, mm) was monthly measured with a flexometer to the nearest mm (Uline Accu-Lock,

H-1766). Survival rate was also determined at 12-month age, at which point 4753 seedlings were alive (Table 4).

**Table 4.** Descriptive statistics for seed traits and survival rate of each provenance of the Mexican spruces studied (* SW = weight of 1000 seeds, SE = standard error, NA = not available; ** PN = number of progenies evaluated).

| Species | Provenance Code | SW (SE) * (g) | PN ** | No. of Seeds Sown | Emerged Seedlings (%) | Survival after 12 Months (%) |
|---|---|---|---|---|---|---|
| *Picea chihuahuana* | Pch-QD | 8.3 (NA) | NA | 93 | 93.5 | 79.2 |
| *P. martinezii* | Pma-AF | 24.4 (0.013) | 20 | 618 | 86.4 | 96.7 |
| | Pma-AA | 22.0 (0.0065) | 30 | 930 | 90.8 | 97.6 |
| | Pma-EB | 23.0 (0.014) | 30 | 930 | 89.5 | 93.5 |
| | Pma-LE | 20.0 (0.00027) | 12 | 375 | 82.9 | 93.7 |
| *P. mexicana* | Pme-EM | 4.9 (0.00072) | 30 | 929 | 95.5 | 87.6 |
| | Pme-LM | 5.1 (0.00072) | 29 | 899 | 91.4 | 90.4 |
| | Pme-EC | 4.9 (0.003) | 28 | 867 | 94.8 | 91.3 |

*2.5. Classification by Cluster Analysis*

To group the eight *Picea* provenances by seedling diameter, height and survival, we used the affinity propagation (AP) clustering technique, with the input preference to the quantile ($q$) = 1 of the input similarities [42] along with the *k*-means clustering algorithm (*k*-means) [43]. We also used the Calinski–Harabasz criterion (CHC) to determine the optimal number of clusters. CHC minimizes the within-cluster sum of squares and maximizes the between-cluster sum of squares. The highest CHC value is related to the optimal set. In this test, the optimal set can be identified by a peak on the linear plot of CHC values [44]. All cluster analyses were implemented using the "*k*-means" and "apcluster" packages [42,43], executed in the *R* free statistical software, version 3.2.3 [45].

*2.6. Pairwise Multiple Comparisons of Mean Rank Sums and Spearman's Correlations*

The median differences in survival rate and growth traits (*D* and *H*) among the three species, among the eight provenances of the three species and among the progenies within each provenance (Table 4) were tested using the non-parametric Tukey–Kramer (Nemenyi) test after corroborating a Tukey-distribution with the PMCMR package [46], implemented in *R*.

We used the Spearman's correlation ($r_s$) test [47] to analyze how survival rate, *D*, *H* and *SW* were related to the climatic and soil variables for each provenance (alpha = 0.025). First, we used the randomForest package in R [48] to detect which variables were most closely correlated with the growth traits. Spearman's correlations and their *p* values were then estimated in *R* [45]. Finally, some linear regression models (including $R^2_{adj}$, its *p* value, slope parameter ($\beta$) and intercept ($\alpha$)) were created using the ggplot [49] package and *lm* function in *R*.

*2.7. Heritability of Picea martinezii and Picea mexicana*

To test whether provenance variation was significant, variance components were obtained using the PROC VARCOMP procedure and the restricted maximum likelihood method in the Statistical Analysis System software (SAS 9.0) [50].

To calculate the heritability and estimate the variance components, we used the MIXED procedure and the REML method, which are considered to be the most appropriate when the data are imbalanced [51], in SAS 9.0. The genetic control of each variable was evaluated using the equations described by [52]:

$$h_i^2 = \sigma_A^2 / (\sigma_f^2 + \sigma_{bf}^2 + \sigma_e^2)$$
$$h_f^2 = \tfrac{1}{4}\sigma_A^2 / \left(\sigma_f^2 + \sigma_{bf}^2 + \tfrac{\sigma_e^2}{b}\right)$$

where: $h_i^2$ = individual heritability; $h_f^2$ = family heritability; $\sigma_A^2$ = additive genetic variance = $3\sigma_f^2$; $\sigma_f^2$ = variance of families; $\sigma_e^2$ = variance of error; $b$ = number of blocks (here $b$ = 21).

A genetic determination coefficient of 3 was used to estimate additive genetic variance under the assumption that intra-class correlation in sibling families obtained by free pollination is 1/3 [53] and not 1/4 as in true half-sibling families [52].

## 3. Results

### 3.1. Provenance Differences in Quantitative Traits

Statistically significant differences between the provenances of the three Mexican spruces (*Picea chihuahuana*, *P. martinezii* and *P. mexicana*) in diameter and height were observed during the first 12 months after germination (Figure 2). However, the growth traits of some provenances were similar (Table 5). Differences in the seedling size of the provenances increased with age (Figure 2).

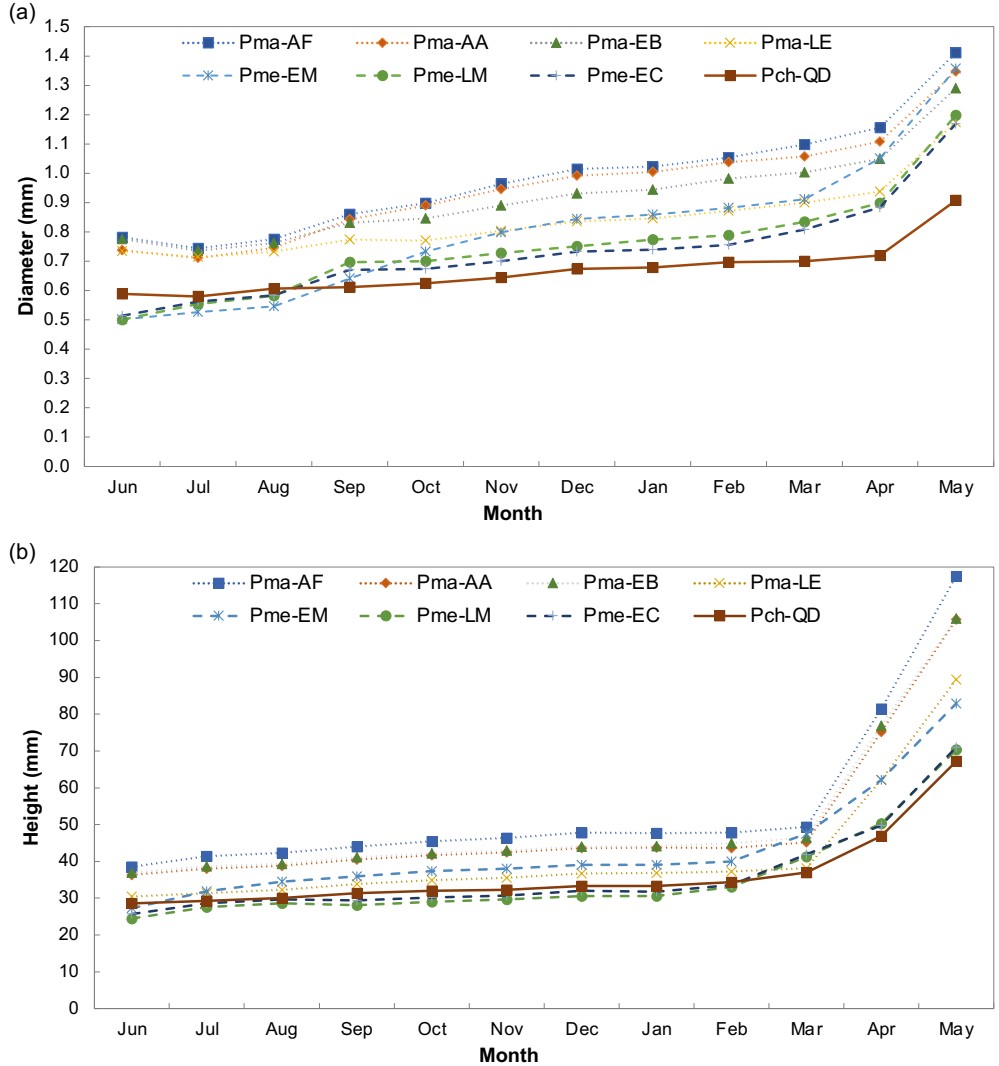

**Figure 2.** Monthly measurements of diameter (**a**) and height (**b**) growth traits of provenances of three Mexican spruces (June 2019—May 2020). *Picea chihuahuana*—only one provenance, Quebrada de los Durán (Pch-QD). *Picea martinezii* (Pma)—four provenances: Agua Fría (Pma-AF), Agua Alardín (Pma-AA), El Butano (Pma-EB) and La Encantada (Pma-LE). *Picea mexicana* (Pme)—three provenances: El Mohinora (Pme-EM), La Marta (Pme-LM) and El Coahuilón (Pme-EC).

**Table 5.** Absolute differences in mean diameter (*D*) and height (*H*) among the eight provenances of three Mexican spruces and respective *p*-values for differences in *H* and *D* (in brackets) (* Pro = provenances of: Pch-QD = *Picea chihuahuana*-Quebrada de los Durán, Pma-AF = *Picea martinezii*-Agua Fría, Pma-AA = *Picea martinezii*-Agua de Alardín, Pma-EB = *Picea martinezii*-El Butano, Pma-LE = *Picea martinezii*-La Encantada, Pme-EM = *Picea mexicana*-El Mohinora, Pme-LM = *Picea mexicana*-La Marta, Pme-EC = *Picea mexicana*-El Coahuilón.

| Pro * | *D* (mm) | *H* (mm) | Pma-AF | Pma-AA | Pma-EB | Pma-LE | Pme-EM | Pme-LM | Pme-EC |
|---|---|---|---|---|---|---|---|---|---|
| Pch-QD | 0.9 | 67 | 5, 50 ($2 \times 10^{-16}$ $7.2 \times 10^{-14}$) | 4, 38 ($2 \times 10^{-16}$ $9.2 \times 10^{-14}$) | 4, 39 ($8.2 \times 10^{-14}$ $1.1 \times 10^{-13}$) | 3, 22 ($1.1 \times 10^{-7}$ $2.2 \times 10^{-5}$) | 5, 15 ($7.2 \times 10^{-14}$ 0.003) | 3, 3 ($1 \times 10^{-10}$ 0.996) | 3, 4 ($1.1 \times 10^{-8}$ 0.996) |
| Pma-AF | 1.4 | 117 | | 1, 12 (0.008 0.001) | 1, 11 ($1.1 \times 10^{-11}$ 0.0006) | 2, 28 ($2 \times 10^{-16}$ $7.9 \times 10^{-14}$) | 1, 23 ($1.9 \times 10^{-5}$ $2 \times 10^{-16}$) | 1, 35 ($2 \times 10^{-16}$ $2 \times 10^{-16}$) | 1, 34 ($2 \times 10^{-16}$ $2 \times 10^{-16}$) |
| Pma-AA | 1.4 | 105 | | | 0, 1 (0.0007 1.000) | 1, 16 ($5.9 \times 10^{-14}$ $1.5 \times 10^{-11}$) | 1, 23 (0.7568 $2 \times 10^{-16}$) | 1, 35 ($1.8 \times 10^{-14}$ $2 \times 10^{-16}$) | 1, 34 ($2 \times 10^{-16}$ $2 \times 10^{-16}$) |
| Pma-EB | 1.3 | 106 | | | | 1, 17 ($2.6 \times 10^{-9}$ $6.8 \times 10^{-11}$) | 1, 24 (0.1625 $2 \times 10^{-16}$) | 1, 36 ($9.2 \times 10^{-11}$ $2 \times 10^{-16}$) | 1, 35 ($6.5 \times 10^{-14}$ $2 \times 10^{-16}$) |
| Pma-LE | 1.2 | 89 | | | | | 2, 7 ($8.2 \times 10^{-14}$ 0.172) | 0, 19 (0.916 $9.5 \times 10^{-14}$) | 0, 18 (1.00 $9.6 \times 10^{-14}$) |
| Pme-EM | 1.4 | 82 | | | | | | 2, 12 ($6.6 \times 10^{-14}$ $5.3 \times 10^{-13}$) | 2, 11 ($1.2 \times 10^{-14}$ $5.8 \times 10^{-13}$) |
| Pme-LM | 1.2 | 70 | | | | | | | 0, 1 (0.681 1.000) |
| Pme-EC | 1.2 | 71 | | | | | | | |

The Calinski–Harabasz criterion (CHC) detected seven clusters based on the seedling survival rate, diameter and height (Figure 3a). Affinity propagation clustering also distinguished seven clusters (i.e., seven provenances) in the three *Picea* species studied, since Pme-LM and Pme-EC are in the same cluster (Figure 3b).

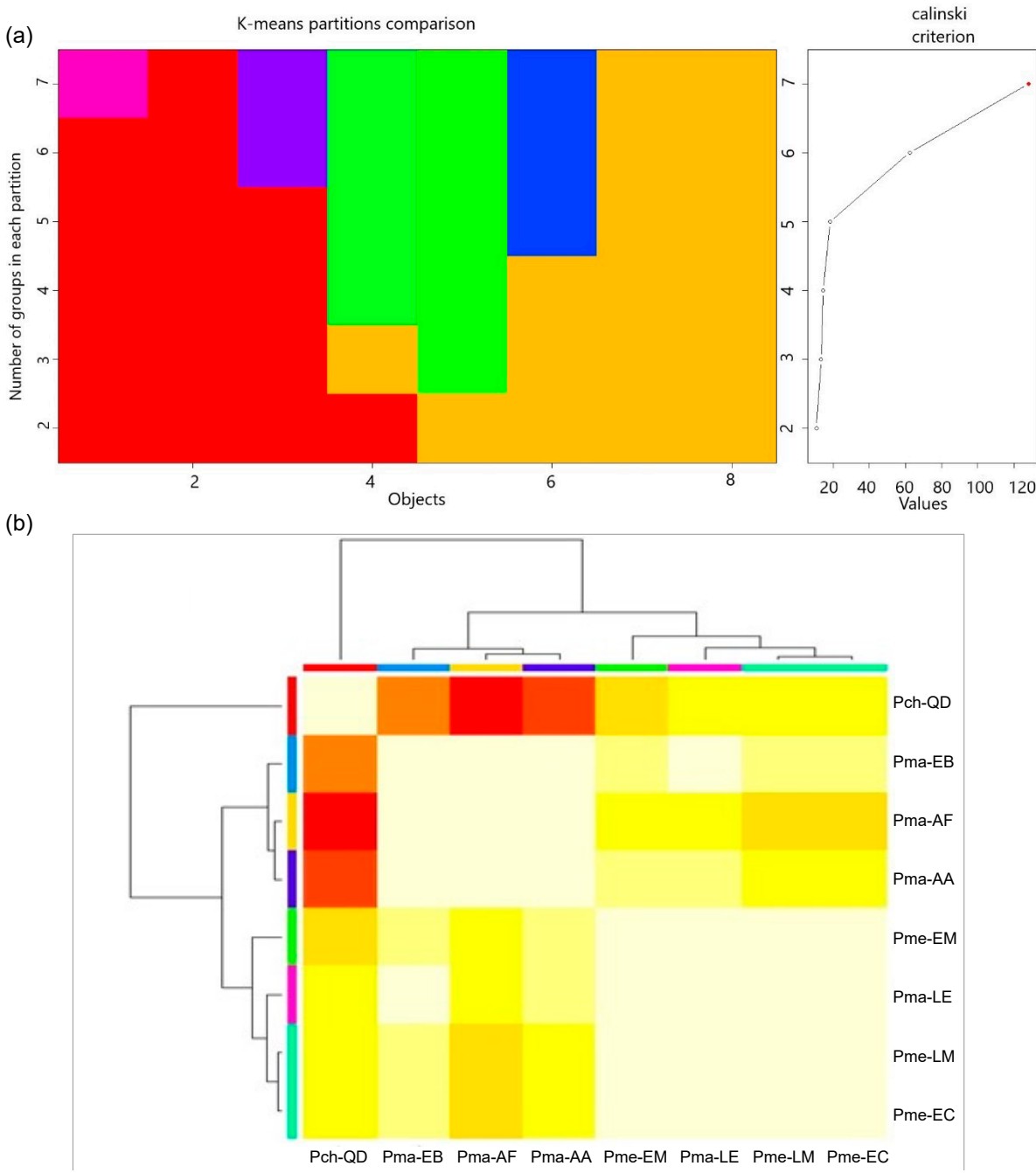

**Figure 3.** Grouping tests for the eight *Picea* provenances by seedling diameter, height and survival. Both the Calinski–Harabasz criterion (**a**) and the affinity propagation clustering technique (*q* = 1) (**b**) distinguished seven clusters. Pch-QD = *P. chihuahuana*—Quebrada de los Durán, Pma-AF = *P. martinezii*—Agua Fría, Pma-AA = *P. martinezii*—Agua Alardín, Pma-EB = *P. martinezii*—El Butano, Pma-LE = *P. martinezii*—La Encantada, Pme-EM = *P. mexicana*—El Mohinora, Pme-LM = *P. mexicana*—La Marta, Pme-EC = *P. mexicana*—El Coahuilón. In (**a**) each color represents a different cluster (group); in (**b**), the upper and left color bars indicate the groups, while the inner red-yellow scaled colors indicate the degree of similarity between provenances (the lighter, the more similar).

The Tukey–Kramer (Nemenyi) tests showed that all Mexican spruce species differed significantly in *H* and all provenances differed significantly in *D* and *H*, except for Pme-LM and Pme-EC (Table 5). However, there were no significant differences in the survival of species or provenances of the Mexican spruces.

### 3.2. Associations between Environmental Variables and Phenotypic Traits

Considering the eight provenances together, significant associations ($p < 0.01$) between *H* and the percentage of organic matter (O.M.) in the soil were observed ($r_s = 0.90$). Strong and significant correlations ($|r_s| > 0.9$, $p < 0.01$) were also observed between the survival rate and mean minimum temperature of the coldest month (Mmin), longitude (Long), length of the frost-free period (Ffp), Julian date of the first autumn freezing (Fday), summer precipitation balance (Smrpb), Julian date of the last spring freezing (Sday) and mean temperature of the coldest month (Mtcm). With regard to seed weight, strongly significant associations ($|r_s| > 0.9$, $p < 0.01$) were detected with elevation (Elev), mean annual temperature (Mat), mean temperature of the warmest month (Mtwm) and degree-days above 5 °C (Dd5) (Table 6). No significant associations between mean seedling diameter and environmental variables were found. Without considering Pch-QD, however, the association between *H* and the percentage of silt in the soil was also very strong and statistically significant ($r_s = -0.99$, $p < 0.01$). Figure 4 shows the linear regressions of seeding height with percentage of organic matter, seed weight with mean annual temperature and seedling survival with mean minimum temperature in the coldest month (°C) of seed origin.

**Table 6.** Spearman correlation ($r_s$) between mean seedling height (H, mm), survival (%) (SU) and weight of 1000 seeds (SW, g) with climatic and soil variables (var) ($\alpha = 0.025$) (* There was strong collinearity of seedlings height, survival and seed weight with respect to some climate variables and also with the percentage of silt (%Silt), organic matter and potassium in the soil ($r_s > 0.9$); O.M. = organic matter (%), Smrp = summer precipitation, Elev = elevation (m), Mmin = mean minimum temperature of the coldest month (°C), Long = longitude (m), Ffp = length of the frost-free period (days), Fday = Julian date of the first autumn freezing (days), Smrpb = summer precipitation balance (ratio), Sday = Julian date of the last spring freezing (days), Mtcm = mean temperature of the coldest month (°C), Mat = mean annual temperature (°C), Mtwm mean temperature of the warmest month (°C) and degree-days above 5 °C.

| | Phenotypes | Variables * | $r_s$ | *p* Value |
|---|---|---|---|---|
| | Height | O.M. | 0.90 | 0.0045 |
| | | Smrp | −0.83 | 0.011 |
| | | Elev | −0.83 | 0.015 |
| | | SW | 0.78 | 0.027 |
| Including Pch—QD | Survival | Mmin | 0.98 | 0.00008 |
| | | Long | −0.96 | 0.0001 |
| | | Ffp | 0.95 | 0.0011 |
| | | Fday | 0.95 | 0.0011 |
| | | Smrpb | −0.92 | 0.0011 |
| | | Sday | −0.93 | 0.0022 |
| | | Mtcm | 0.93 | 0.0022 |
| | | Mmindd0 | −0.88 | 0.0022 |
| | | Height | 0.81 | 0.021 |

**Table 6.** *Cont.*

|  | Phenotypes | Variables * | $r_s$ | *p* Value |
|---|---|---|---|---|
| Including Pch—QD | Seed weight | Elev | −0.90 | 0.004 |
| | | Mat | 0.90 | 0.0045 |
| | | Smrp | −0.87 | 0.0045 |
| | | Mtwm | 0.90 | 0.0045 |
| | | Dd5 | 0.90 | 0.0045 |
| | | Map | −0.85 | 0.0072 |
| | | Smrsprpb | −0.84 | 0.0093 |
| | | Smrpb | −0.84 | 0.0093 |
| | | Height | 0.72 | 0.027 |
| Excluding Pch—QD | Height | %Silt | −0.99 | 0.00001 |
| | | Elev | −0.96 | 0.0027 |
| | | SW | 0.89 | 0.012 |
| | | Mmax | 0.85 | 0.016 |
| | | Potassium | 0.85 | 0.016 |
| | | O.M | 0.86 | 0.023 |
| | Seed weight | Smrp | −0.954 | 0.0008 |
| | | Smrpb | −0.900 | 0.0056 |
| | | Smrsprpb | −0.928 | 0.0067 |
| | | %Silt | −0.86 | 0.011 |
| | | Map | −0.89 | 0.012 |
| | | Mmax | 0.85 | 0.016 |
| | | Mtcm | 0.86 | 0.023 |
| | | Mat | 0.86 | 0.023 |
| | | Elev | −0.86 | 0.023 |
| | | D100 | −0.86 | 0.023 |

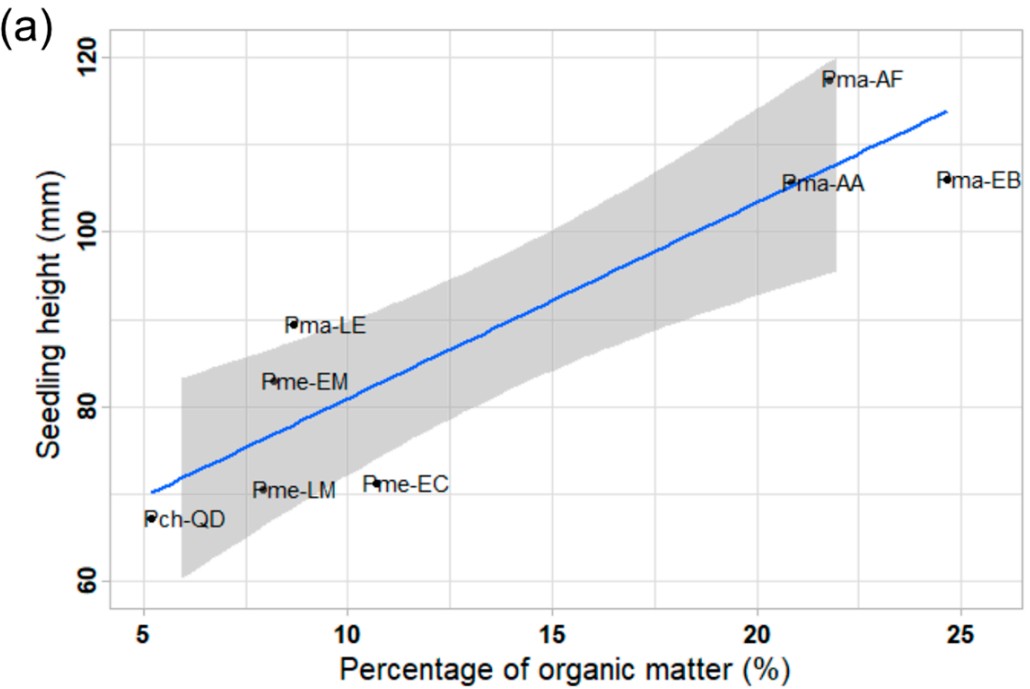

**Figure 4.** *Cont.*

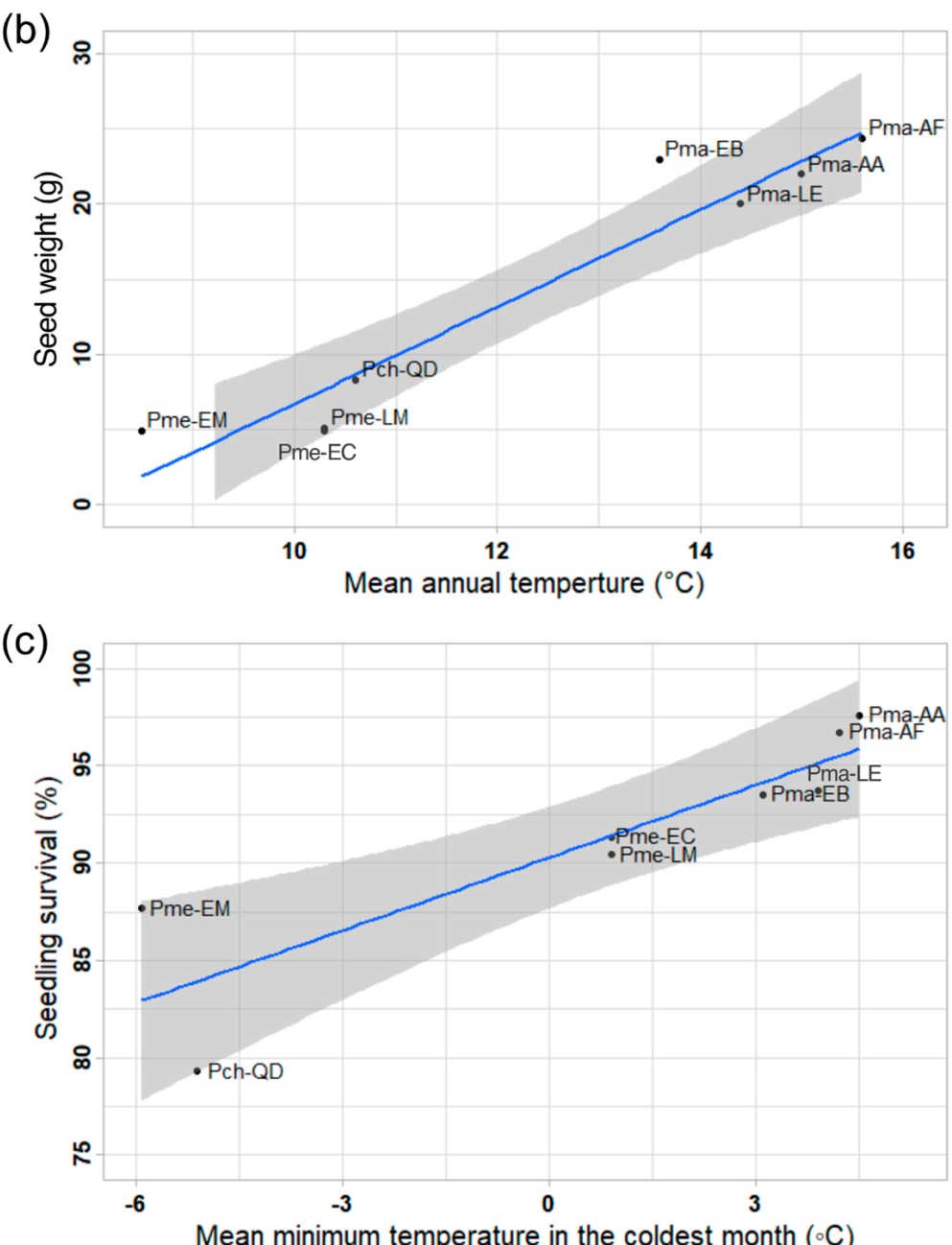

**Figure 4.** Linear regression models of (**a**) seedling height with percentage of organic matter ($R^2_{adj}$ = 0.77, *p* value = 0.0026, slope parameter ($\beta$) = 2.24, intercept = 58.54); (**b**) seed weight with mean annual temperature ($R^2_{adj}$ = 0.90, *p* value = 0.0002, slope parameter ($\beta$) = 3.23, intercept= −25.68); and (**c**) seedling survival with mean minimum temperature of the coldest month (°C) ($R^2_{adj}$ = 0.75, *p* value = 0.0035, slope parameter ($\beta$) = 1.25, intercept = 90.27) of seed origin; blue lines = mean values and grey area = 95% confidence level intervals for predictions; Provenances of Pch-QD = *Picea chihuahuana*—Quebrada de los Durán, Pma-AF = *P. martinezii*—Agua Fría, Pma-AA = *P. martinezii*—Agua Alardín, Pma-EB = *P. martinezii*—El Butano, Pma-LE = *P. martinezii*—La Encantada, Pme-EM = *P. mexicana*—El Mohinora, Pme-LM = *Picea mexicana*—La Marta, Pme-EC = *P. mexicana*—El Coahuilón.

### 3.3. Heritability (Within-Provenance) of Picea martinezii and P. mexicana Growth Traits

The medium to high within-provenance heritability values of the growth traits *D* and *H* and notable percentage of provenance variation relative to total variance for *Picea martinezii* and *P. mexicana* seedlings are shown in Table 7.

**Table 7.** Heritability (within-provenance) of seedling diameter and height (growth traits) in *Picea martinezii* and *P. mexicana*. Individual ($h_i^2$) and family ($h_f^2$) heritability ± standard error are shown (* relative to total variance).

| Species | Seedling Growth Trait | Percentage of Provenance-Related Variation * | $h^2{}_i$ | $h^2{}_f$ |
|---|---|---|---|---|
| *Picea martinezii* | Diameter | 14.5 | 0.19 ± 0.55 | 0.64 |
| | Height | 13.4 | 0.21 ± 0.35 | 0.65 |
| *Picea mexicana* | Diameter | 8.1 | 0.47 ± 0.72 | 0.83 |
| | Height | 14.9 | 0.55 ± 0.60 | 0.85 |

## 4. Discussion

The study findings show significant (phenotypic) differences between the three spruce species considered, as well as between the provenances within the species, in seedling growth traits under the same environmental conditions (i.e., greenhouse) (Table 5, Figure 3). The differences in provenances could be caused by the high degree of spatial, ecological and temporal variation and possibly by the pre- or postzygotic isolation of these provenances and the divergent natural selection in heterogeneous environments, e.g., large differences in elevation, temperature and some soil conditions [20,28]. This selection has resulted in locally adapted plant genotypes [54]. It can be assumed that the observed growth differences will increase over time (Figure 2).

The strong correlations between the measured traits (height, survival rate and weight of 1000 seeds) with several climatic variables from the areas of seed origin indicate some degree of adaptation to local conditions that determines the variation in seedling growth in the spruce provenances considered. The prevailing climatic conditions in natural locations of the best-performing spruce (*P. martinezii*), particularly for the two best-performing provenances (AA and AF) (Table 5), were closer to climatic greenhouse conditions and, therefore, these provenances were probably best adapted (Table 6 and Figure 4). In addition, AA and AF (like all *P. martinezii* provenances) had a particularly high seed weight (*SW*) (Table 4) (see below), which also strongly promoted their faster initial growth in diameter and height (*H*) (Figure 2). *SW* is both a part of maternal effects [55] and a largely genetically controlled quantitative trait [17,55]. The high level of local adaptation is consistent with the high degree of genetic differentiation among the three spruce species and their provenances [18,22–24,30].

Other studies have also shown that adaptation is directly related to the climate gradient in the area where the species are distributed. For example, Rehfeldt [56] showed the adaptation of *Picea engelmannii* populations to heterogeneous environments. Oleksyn et al. [57] presented common garden evidence for elevational ecotypes and the cold adaptation of *Picea abies*. Islam and Macdonald [58] showed ecophysiological adaptations of *Picea mariana* and *Larix laricina* seedlings to flooding. Sáenz-Romero et al. [34] reported elevational-related genetic variation among *Pinus oocarpa* populations. Andersen et al. [59] observed differences in growth traits (number of secondary branches and seedling height), which were caused by differences in abiotic factors, in nine provenances of *Abies guatemalensis*. Castellanos-Acuña et al. [60] showed that the mean temperature of the coldest month and an aridity index are strongly related to the genetic adaptation of Mexican tree species. In addition, Konôpková et al. [61] concluded that the Central European provenances of *Abies alba* from higher elevations, associated with wetter and colder conditions, demonstrated the greatest photosynthetic production and were less sensitive to moderate heat and dryness. Petrík et al. [62] found, in a study of 20 provenances of *Fagus sylvatica* in Europe, that saplings from hotter, drier environments showed the greatest growth in height, as we found in our study (Table 6).

Strong associations between seedling height and soil variables of seed origin were also detected (Table 6, Figure 4). Similarly, Pickles et al. [63] reported that the height of Douglas-fir seedlings was optimized after the transfer of seeds to drier soils, whereas

survival was improved when elevation transfer was minimized, and they concluded that soil biota interacts with biogeography to influence plant ranges. Macel et al. [54] found no evidence of adaptation to climate in the legume *Lotus corniculatus*, and only differences in number of fruits regarding soil origin, showing a weak indication of adaptation to soil conditions.

We observed strong associations between *SW* and *H* in relation to climatic and soil variables (Table 6), probably due to control by quantitative trait loci [17]. Similarly, the results of a provenance trial under nursery conditions with Ethiopian *Cordia africana* showed that *SW* was significantly and positively correlated with elevation and negatively correlated with mean annual temperature of seed origin [64]. Leal-Sáenz et al. [65] reported that *Pinus strobiformis* populations in more humid and warmer climates and growing in dense, shady forest stands had heavier seeds because larger seeds could provide greater resources for germination.

The medium-to-high within-provenance heritability values (Table 7), representing the proportion of phenotypic variation attributable to genetic variability within populations, also support the hypothesis of adaptation to local conditions for each provenance under study [20]. However, phenotypic plasticity, which enables species to cope with different environmental scenarios, can explain the complementary part of the growth differences [66,67]. Finally, environmental variation does not necessarily lead to adaptive variation and can also result in non-adaptive phenotypic plasticity [68]. According to the results obtained, the adaptation of provenances of the Mexican spruces to local conditions is more likely. Hence, the use of combinations of germplasm from different provenances for both in situ and ex situ activities should be carried out with caution. Thus, the conservation of current genetic pools should be prioritized, whereas the exchange of germplasm among provenances should be reserved for the genetic rescue of populations with a high extinction risk.

## 5. Conclusions

Applying the concept of trans-specific traits to the three Mexican spruce species makes local adaptation more visible than these traits only at the species level, probably due to the stronger isolation mechanisms and greater genetic distances among these species, and the different species' niches.

The study findings indicate the need to consider environmental conditions for future conservation actions and to analyze the relationships between the variation in environmental factors (such as precipitation and edaphic traits) and the growth patterns of tree species. This would enable the identification of favorable forest conditions and reforestation with seedlings of suitable species and origins for each site, considering the environmental setting. By considering the predicted future environmental conditions for each site to be reforested, increased survival rates can also be achieved by reducing the mortality due to the decoupling of plant requirements from the climates and soil in which they have evolved over long periods. Thus, reforestation programs for these species with isolated and fragmented populations should be more successful when seedlings from one provenance are planted in environmental conditions similar to those of the same provenance.

However, the very small ranges and habitats of these three endemic species and the very small sizes of provenances, along with the marked local adaptation detected, make both germplasm exchange among provenances and the ex situ conservation of genetic variation very difficult.

**Author Contributions:** Conceptualization, C.W.; methodology, C.W.; software, J.M.T.-V.; validation, C.W., J.C.H.-D. and A.C.-P.; formal analysis, J.M.T.-V. and J.L.-U.; investigation, J.M.T.-V.; resources, E.M.-M.; data curation, J.M.T.-V. and S.S.-G.; writing—original draft preparation, C.W.; writing—review and editing, J.L.-U., J.M.T.-V., J.C.H.-D., A.C.-P. and E.M.-M.; visualization, J.C.H.-D.; supervision, C.W.; project administration, C.W. and S.S.-G.; funding acquisition, C.W. All authors have read and agreed to the published version of the manuscript.

**Funding:** This material is based on work supported by the Consejo Nacional de Ciencia y Tecnología of México (CONACYT) and the Comisión Nacional Forestal (CONAFOR). Moreover, CONACYT provided financial support to José Marcos Torres-Valverde for his master's studies (CVU No. 1004398).

**Institutional Review Board Statement:** Not applicable.

**Informed Consent Statement:** Not applicable.

**Data Availability Statement:** Not applicable.

**Acknowledgments:** We thank Angélica Rocio Espinoza Delgado, Ana Maria Ibarra, Emma Lizbeth Espinoza Delgado, Ricardo Silas Sánchez-Hernández and Oscar Alfredo Díaz-Carrillo for assistance with the fieldwork.

**Conflicts of Interest:** The authors declare no conflict of interest.

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
