# Peer review of "Adaptive Trait Variation in Seedlings of Rare Endemic Mexican Spruce Provenances under Nursery Conditions"

_forests, doi:10.3390/f14040790_

Round 1

Reviewer 1 Report (Previous Reviewer 3)

In my opinion, with the corrections made, the text can be published.

Not all my comments were understood by the authors. For example, when presenting an ordinal regression equation he often performs The F-Test of overall significance in regression is a test of whether or not your linear regression model provides a better fit to a dataset than a model with no predictor variables. Linear regression needs the relationship between the independent and dependent variables to be linear. However, this is not a key issue in interpreting the results.

In my opinion, with the corrections made, the text can be published. Almost all my comments have been taken into account.

Not all my remarks were understood by the authors. For example, when presenting an ordinal regression equation he often performs The F-Test of overall significance in regression is a test of whether or not your linear regression model provides a better fit to a dataset than a model with no predictor variables. Linear regression needs the relationship between the independent and dependent variables to be linear. However, this is not a key issue in interpreting the results.

In my opinion, the work is interesting and meets the standards of a scientific paper

Author Response

Comments and Suggestions for Authors

In my opinion, with the corrections made, the text can be published. Almost all my comments have been taken into account.

Not all my remarks were understood by the authors. For example, when presenting an ordinal regression equation he often performs The F-Test of overall significance in regression is a test of whether or not your linear regression model provides a better fit to a dataset than a model with no predictor variables. Linear regression needs the relationship between the independent and dependent variables to be linear. However, this is not a key issue in interpreting the results.

We added in Figure 4: “… (R2adj = 0.77, p value = 0.0026, slope parameter (β) = 2.24, intercept = 58.54);…(R2adj = 0.90, p value = 0.0002, slope parameter (β) = 3.23, intercept= -25.68); … (R2adj = 0.75, p value = 0.0035, slope parameter (β) = 1.25, intercept = 90.27) …”.

In my opinion, the work is interesting and meets the standards of a scientific paper

Many thanks!

Reviewer 2 Report (New Reviewer)

The manuscript is interesting to publish in this journal. Following minor changes can made

Line 27-28: rewrite in the abstract

Lines 57-59: please remove links from introduction

Remove headings from discussion

Remove citations from conclusion

Add reference in discussion from last five years

Author Response

Comments and Suggestions for Authors

The manuscript is interesting to publish in this journal. Following minor changes can made

Line 27-28: rewrite in the abstract

We have rewritten this text, now:  “All Mexican spruce species differed significantly in H, and all provenances studied were significantly different in D and H, except for two neighbouring provenances of P. mexicana. Very strong, significant correlations (up to R2 = 0.96) were found between H, survival rate and SW with respect to environmental factors of provenance/seed origin.”

Lines 57-59: please remove links from introduction

done

Remove headings from discussion

done

Remove citations from conclusion

done

Add reference in discussion from last five years

done

Konôpková, A.; Pšidová, E.; Kurjak, D.; Stojnić, S.; Petrík, P.; Fleischer Jr, P.; Kučerová, J.; Ježík, M.; Petek, A.; Gömöry, D.; Kmeť, J., Photosynthetic performance of silver fir (Abies alba) of different origins under suboptimal growing conditions. Functional Plant Biology 2020, 47(11), 1007-1018.

Petrík, P.; Grote, R.; Gömöry, D.; Kurjak, D.; Petek-Petrik, A.; Lamarque, L.J.; Sliacka Konôpková, A.; Mukarram, M.; Debta, H.; Fleischer Jr, P.. The Role of Provenance for the Projected Growth of Juvenile European Beech under Climate Change. Forests 2022, 14(1), 26.

We added (lines 375-380): “In addition, Konôpková et al. [62] concluded that the Central European provenances of Abies alba from higher elevations, associated with wetter and colder conditions, demonstrated the greatest photosynthetic production and were less sensitive to moderate heat and dryness. Petrík et al. [63] found in a study of 20 provenances of Fagus sylvatica in Europe, that saplings from hotter, drier environments showed the greatest growth in height, as we found in our study (Table 6).”

Reviewer 3 Report (New Reviewer)

The paper by Wehenkel et al., titled: “Adaptive trait variation in seedlings of rare endemic Mexican spruce provenances under nursery conditions”, is original work, fitting to the scope of Forests journal.

The paper explored seed weight, seedlings survival and growth variability among provenances and species of Mexican spruces. Authors further analysed the heritability and impact of original climate on these characteristics. The results provide practical information for assisted migration strategies of adaptive forestry under climate change. The patterns between original climate and expressed phenotypes improve the understanding of spruces ecogenetics. The value of the paper is also higher because as it is first report of local adaptation of the three spruce species used in this study. The paper is overall well composed and easy to read. Abstract is concise and informative. Introduction gives overview of the problematic, but could be improved. Authors could expand the significance of assisted migration and provenance specific afforestation strategies more. Materials and methods are well described, statistical analyses are adequate. Visual presentation of results is clear and informative. The discussion can be improved with more extensive comparison with other provenance studies. Conclusion is very well written. I am looking forward to future provenance studies from the authors.

I suggest minor revision of the paper.

My comments:

Abstract

Line 29: Maybe you could specify to environmental factors of provenance/seed origin.

Introduction

Missing references from key-figures of the provenance field: Prof. Gömöry, Prof. Mátyás, Prof. Kramer, Prof. Hellmann. I think mentioning some of the key studies focused on provenance research and assisted migration can give better overview for the readers.

Materials and methods:

Line 131: The link is not working for me.

Table 2: You could add the time period 1961-1990 also in the table description.

Line 148: When were the soil samples collected? Please add the info to the text.

Discussion

I think that the length of the discussion is not adequate to the presented results. See other provenance studies analyzing norway spruce provenance responses:

Schmidtling 1994, https://doi.org/10.1093/treephys/14.7-8-9.805

Klisz et al. 2019, https://doi.org/10.3389/fpls.2019.00306

Line 363: I recommend expanding of this chapter and providing more examples for the climatic adaptation in tree provenances: altitude/precipitation/temperature adaptation of photosynthetic performance of Abies alba: Konôpkova et al. 2020, https://doi.org/10.1071/FP20040; aridity/temeprature adaptation of growth in Fagus sylvatica: Petrík et al. 2022, https://doi.org/10.3390/f14010026.

Author Response

Reviewer 3

I suggest minor revision of the paper.

My comments:

Abstract

Line 29: Maybe you could specify to environmental factors of provenance/seed origin.

done

Introduction

Missing references from key-figures of the provenance field: Prof. Gömöry, Prof. Mátyás, Prof. Kramer, Prof. Hellmann. I think mentioning some of the key studies focused on provenance research and assisted migration can give better overview for the readers.

We added following references in the Introduction: 

Matyas, C. Climatic adaptation of trees: rediscovering provenance tests. Euphytica 1996, 92, 45-54.

Kramer, K.; Degen, B.; Buschbom, J.; Hickler, T.; Thuiller, W.; Sykes, M.T.; de Winter, W. Modelling exploration of the future of European beech (Fagus sylvatica L.) under climate change—range, abundance, genetic diversity and adaptive response. Forest Ecology and Management 2010, 259(11), 2213-2222.

Gömöry, D.; Longauer, R.; Hlásny, T.; Pacalaj, M.; Strmeň, S.; Krajmerová, D. Adaptation to common optimum in different populations of Norway spruce (Picea abies Karst.). European Journal of Forest Research 2012, 131, 401-411.

Materials and methods:

Line 131: The link is not working for me.

We changed the linke, now: https://charcoal2.cnre.vt.edu/climate/

Table 2: You could add the time period 1961-1990 also in the table description.

done

Line 148: When were the soil samples collected? Please add the info to the text.

done

Discussion

I think that the length of the discussion is not adequate to the presented results. See other provenance studies analyzing norway spruce provenance responses:

Schmidtling 1994, https://doi.org/10.1093/treephys/14.7-8-9.805

Klisz et al. 2019, https://doi.org/10.3389/fpls.2019.00306

Line 363: I recommend expanding of this chapter and providing more examples for the climatic adaptation in tree provenances: altitude/precipitation/temperature adaptation of photosynthetic performance of Abies alba: Konôpkova et al. 2020, https://doi.org/10.1071/FP20040; aridity/temeprature adaptation of growth in Fagus sylvatica: Petrík et al. 2022, https://doi.org/10.3390/f14010026.

We added (lines 360-363): “In addition, AA and AF (like all P. martinezii provenances) had particularly high seed weight (SW) (Table 4) (see below), which also strongly promoted their faster initial growth in diameter (D) and height (H) (Figure 2). SW is both a part of maternal effects [55] and a largely genetically controlled quantitative trait [17, 55].”

and

(lines 375-380): “In addition, Konôpková et al. [62] concluded that the Central European provenances of Abies alba from higher elevations, associated with wetter and colder conditions, demonstrated the greatest photosynthetic production and were less sensitive to moderate heat and dryness. Petrík et al. [63] found in a study of 20 provenances of Fagus sylvatica in Europe, that saplings from hotter, drier environments showed the greatest growth in height, as we found in our study (Table 6).”

This manuscript is a resubmission of an earlier submission. The following is a list of the peer review reports and author responses from that submission.

Round 1

Reviewer 1 Report

By comparing the phenotypic differences of eight Picea provenances belong to 3 Picea species, the authors want to illustrate the variation of adaptive traits of different Mexican spruce. Association and cluster analysis with climate and soil variables were also performed. The authors have done a lot of works, but the research still remain some problems in experimental design, data analysis and writing, as follows:

1.     Abstract.  In this section, you just listed your results. Sum up your results and tell us the conclusion. A clear and concise abstract is better.

2.     Background. Some description logic is confusing. In addition, according to the manuscript, In addition to geographical distribution, the author should also briefly introduce the breeding research of three rare spruce species

3.     Materials: According to description in Abstract, 5641seedlings were used in this study, but in table 1, a total of 180 trees were collected. And what’s the population size?

4.     Why just one trees of Picea chihuahuana was collected. Is it representative?

5.     Almost all the figures and tables do not conform to the specifications and need to be redrawn. Why are there two blue lines in Table 3? A-B in Figure 4 is not aligned with C-D

6.     There are many grammatical errors in the manuscript, the logics of some sentences are confusing and does not meet the writing standards. English may need to be checked throughout the text by a native English speaker.

Reviewer 2 Report

The  paper has following preblem:1) Inappropriate keyword selection, Picea and Mexican spruces only need one, proxy is not suitable for keywords.

(2) Provenance refers to the source or origin of a group of seeds or seedlings in the distribution area of the same tree species. Picea chihuahuana and Picea martinezii in the paper are not the same species and have morphological characteristics and genetic differences, so it is meaningless to conduct provenance traits, genetic variation analysis and cluster analysis on themPicea chihuahuana has only one provenance, did not meet the provenance test requirements.

3The units of calcium, magnesiumphosphorus and others in Table 3 are ppm, which is recommended to be changed into international units.

(4) In Figure 2, it is inappropriate to use (diameter x height) to reflect the growth of seedling diameter and height of different provenances, which cannot reflect the difference of diameter and height well.

(5) There is only one provenance and one soil sample for PCH-QD, how to analyze their correlation.

(5) Line 334 Picea martinezii and P. mexicana should be changed to Picea martinezii and P. chihuahuana.

It is suggested that the author remove Picea chihuahuana, re-analyze and rewrite .

Reviewer 3 Report

Describing the diversity of relict species is scientifically important. Also, any action taken to preserve them in the wild and ex-situ is important for many reasons. In my opinion, therefore, the topic of the paper is aptly chosen and worthy of publication. There is, in my opinion, an unnecessary reference to forest tree selection breeding and provenance research, as endemic species are not in themselves of economic importance. I would therefore suggest redirecting the discussion towards characterizing the diversity and conservation of genetic variation in rare (endemic) spruce species. 

In this context, the objectives of the work mentioned at the end of the introduction could also be redefined. The objectives as they stand are unambitious and boil down to the task of "estimating the phenotypic variation in seedlings of the three Mexican spruces of different provenances". 

As for the methodology, my comments are as follows:

Table 4. Provenance code wrapping only a two-letter abbreviation is hardly informative. Therefore, the authors use fuller designations on maps and figures that include the species. I propose to use the same designations in Table 4. 

SW - SW = 100-seed weight typically this parameter is given as the average weight of 1000 seeds in (g). I propose to give in the same way in the table not the weight of 100 but 1000 seeds. For the first sample, all available seeds were weighed. This can be explained by endemic and rare species. 

Heritability - I understand that author can take a different ratio (genetic determination coefficient) not 1/4 but 1/3. I do not understand why the equation from line 182 has not changed. It should be 1/3 there. For heritability parameters, please provide the standard error. SAS calculates this parameter automatically, or you can calculate it separately using one of the recommended methods. 

Comments on the results:

Table 5 It is not good to mix species and provenance variability. Usually, the species variability is higher. What is the purpose of such a comparison? Please address this in the discussion. Otherwise, it would be necessary to reorganize all results and discuss them separately for species and then for provenance

Table 6 The result obtained (in my opinion) is mainly due to differences in geography and elevation.  Locations are related to provenance, but in principle, we are not assessing provenance here because differences due to mountain location come to the fore (for example Pma-AF 1.820 - Pme-EC 3.528) this only makes sense for characterization and not for analysis of differences and correlations.  Please refer to this in the discussion. Fig. 4 and Tab 6 address the same issue of the correlation of traits between climatic and soil variables and seedling diameter, height, seed weight, and survival rate. This creates confusion and it is not clearly described why the authors present the data in this way.  I propose to clarify this and the selected relationships illustrated graphically (fig 4) describe what type of corr was presented. Is this a duplication of information from Table 6? 

(Example Height - %Silt rs [H x var] -0.99 p-value 0.00001 and Figure 4. B)

The discussion in my opinion is a weakness of this paper.  It could have been more directed at the interchangeability of endemic species and the conservation of variation. The correlations compared are not the most important here. The recommendation "we recommend to verify these 358 results with Environmental Association Analysis (EAA) which assesses whether a certain 359 allele or genotype of a locus is linked with a certain environmental condition, while con-360 trolling for the neutral genetic structure" does not follow from the study. The authors have not studied genetic variation at the molecular level, so these are mere generalisations. In my opinion, they should be avoided.  

In my opinion, the whole text does not need many corrections it is interesting and interestingly presented.